# Position: We Need A Unified Definition of Hallucination (It's The World Model, Stupid!)

**Emmy Liu** [1 2]  **Varun Gangal** [1 3]  **Chelsea Zou** [1 4]  **Michael Yu** [1 5]  **Xiaoqi Huang** [1 5]  **Alex Chang** [1 5]  **Zhuofu Tao** [1 5]
**Karan Singh** [1 4]  **Sachin Kumar** [1 6]  **Steven Y. Feng** [1 4]

## Abstract

Despite numerous attempts at mitigation since the inception of language models, hallucinations remain a persistent problem even in today's frontier LLMs. Why is this? We review existing definitions of hallucination and fold them into a single, unified definition wherein prior definitions are subsumed. This position paper argues that *hallucination can be unified by defining it as simply inaccurate (internal) world modeling*, in a form where it is observable to the user. For example, stating a fact which contradicts a knowledge base OR producing a summary which contradicts the source. By varying the reference world model and conflict policy, our framework unifies prior definitions. We argue that this unified view is useful because it forces evaluations to clarify their assumed reference "world", distinguishes true hallucinations from planning or reward errors, and provides a common language for comparison across benchmarks and discussion of mitigation strategies. Building on this definition, we also connect our framework to HALLUWORLD (Liu et al., 2026), a complementary benchmark that instantiates fully specified reference world models for stress-testing model hallucinations.

## 1. Introduction

Suppose a language model is given this passage in context: *"Sherlock Holmes lives at 221B Baker Street in London"* and is asked the question *"Where does Sherlock Holmes live?"*

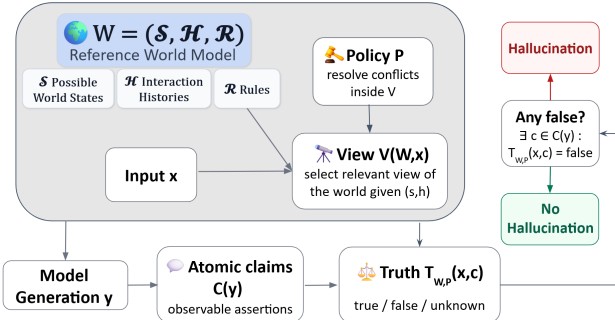

*Figure 1.* Hallucination as inaccurate (internal) world modeling

It responds with *"Sherlock Holmes is a fictional character and has no real address"*. Is this statement considered a hallucination? A summarization researcher might say yes, as the model contradicted the source. An open-domain QA researcher might say no, as it is true that Sherlock Holmes has no real world address. Hallucination, as a term, has evolved since its first introduction, and now means different things in different contexts. This fragmentation of definitions, tailored to task-specific assumptions about what constitutes truth, makes it difficult to answer basic questions about hallucination such as: *Can LLMs ever be expected to stop hallucinating?* or *Can we guarantee that hallucinations will happen a predictable percentage of the time?* If we report that a technique "reduces hallucinations by 40%", what is this actually measuring, and can we expect these improvements to transfer across contexts? These are not just philosophical questions, but have real impact on research directions and the public's trust in deployed systems.

We argue that **existing definitions of hallucination can be unified as inaccurate world modeling that is observable to the user through model outputs**. More explicitly, we argue that every setting implicitly assumes (1) a reference world model encoding what is true, (2) a view function specifying what information the model can observe, and (3) a conflict policy determining how contradictions resolve. A model hallucinates when its outputs imply claims that are false according to the reference world model and conflict policy. Prior definitions simply make different choices for these components, but all share this underlying structure.

Our benchmark: github.com/DegenAI-Labs/HalluWorld [1]DegenAI Labs [2]Carnegie Mellon University [3]Patronus AI [4]Stanford University [5]Independent Researcher [6]The Ohio State University. Correspondence to: Emmy Liu <emmy@cmu.edu>, Varun Gangal <vgtomahawk@gmail.com>, Steven Y. Feng <syfeng@stanford.edu>.

*Proceedings of the $43^{rd}$ International Conference on Machine Learning*, Seoul, South Korea. PMLR 306, 2026. Copyright 2026 by the author(s).

We further argue that making this structure explicit has practical benefits. First, it forces benchmark designers to be more explicit about their assumptions, enabling better comparison across existing benchmarks. It also distinguishes hallucination from other errors that can arise even with a correct world model, and provides a common language for why certain mitigations help in some settings but not others. Lastly, it enables the creation of a class of larger-scale and practically useful benchmarks by using synthetic environments where the reference world is fully specified (game environments, simulated databases, or structured worlds). By having a clear definition, we can more easily generate evaluation instances where hallucination labels are fully determined rather than requiring human annotation.

We start by reviewing definitions of hallucination and how they have evolved over time (§2) before formalizing our framework and unifying existing definitions (§3). We argue in §4 for the benefits of unification, and outline in §5 how our framework enables the creation of scalable benchmarks for advancing research on hallucination. Lastly, we discuss some alternative views in §6 and a call to action in §8. This position paper is complemented by HALLUWORLD (Liu et al., 2026), a benchmark suite that operationalizes our framework in synthetic environments with fully specified reference worlds.

## 2. A History of Hallucination Definitions

### 2.1. Generations Ungrounded in the Source

The original notion of hallucination was introduced in neural machine translation work, and referred to translations which were unrelated to the source text (Lee et al., 2019). This original definition focused on perturbations to an input sentence which would cause the translation to become ungrounded in the input. There is a notable body of follow-up work (Raunak et al., 2021; Guerreiro et al., 2023; Dale et al., 2023; Xu et al., 2023) that deals with analysis, benchmark creation, or mitigation methods based on this original notion, and was extended to tasks like surface realization (Nie et al., 2019) and table-to-text generation (Parikh et al., 2020).

Abstractive summarization introduced a closely related notion. A summary is said to contain hallucinations if it contains spans that are not supported by the input document, even if they might be factually correct in real world terms (Maynez et al., 2020). Further, the distinction between intrinsic and extrinsic hallucinations became standard (Ji et al., 2023): *intrinsic* hallucinations directly contradict the source, while *extrinsic* hallucinations are statements that cannot be verified from the source but are not necessarily false.

### 2.2. From Source Groundness to "Unfactual" Content

As language models have become more capable, the meaning of hallucination has grown to encompass not just generations that are ungrounded in the input, but also generally unfactual outputs. In many current works, hallucination simply refers to fluent content that is factually wrong with respect to some (often implicit) notion of world knowledge. Similar failures arise in concept-to-text generation (Lin et al., 2020; Feng et al., 2021), where models produce fluent but commonsense-violating outputs. The focus is less on faithfulness to a particular input document, and more on consistency with "the world".

Recent surveys such as Ji et al. (2023) reflect this shift by defining hallucination as "generated content that is nonsensical or unfaithful to the provided source content", bringing both plausibility with respect to contextual semantics and factual unfaithfulness under a single umbrella. This definition still foregrounds a provided source when one exists, but in practice the survey and many later works broaden "source" to include external knowledge bases or general world knowledge. Works such as Chen et al. (2024); Rashad et al. (2024) also collect corpora around variations of the generally unfactual vs. factual definition of hallucination, referring to these as "fact-conflicting" and "fact-level", respectively.

### 2.3. Fine-Grained Hallucination Taxonomies

Recent work has moved toward fine-grained classification and detection of hallucinations in long-form generation. FavaBench (Mishra et al., 2024) introduces a taxonomy of hallucination types with span-level annotations in information-seeking settings, training retrieval-augmented models to detect and edit such errors. Despite this increased granularity, hallucinations are still defined relative to a fixed reference source: spans are considered contradictory or unverifiable depending on their relationship to externally retrieved evidence. Hence, these approaches primarily measure factual precision with respect to a chosen corpus, rather than broader failures of a model's internal world representation.

Likewise, HALoGEN (Ravichander et al., 2025) evaluates hallucinations across multiple knowledge-intensive domains by decomposing model outputs into verifiable atomic statements and checking them against external tools or knowledge bases. While the benchmark distinguishes hallucinations based on whether incorrect facts were present during pretraining, hallucinations are identified by comparison to an externally defined notion of truth. HalluLens (Bang et al., 2025) proposes three benchmarks to evaluate *extrinsic* hallucination with respect to the model's pretraining data (fixed and static). Hence, these benchmarks implicitly frame hallucination evaluation as a static, single-turn judgment against a fixed reference(s), rather than as a failure of a model's internal world representation or decision-making over time.

## 2.4. Agentic Hallucinations

Early definitions of hallucination focus on single-turn model outputs. With the rise of agentic LLMs, hallucination becomes an action-level phenomenon: models act across multiple turns and interact with environments. MIRAGE-Bench (Zhang et al., 2025d) formalizes this shift by evaluating hallucinations as unfaithful or inappropriate actions conditioned on task instructions, execution history, or environment observations, rather than isolated QA errors. In parallel, mitigation methods—particularly in code domains—have evolved toward improving an LLM's awareness of its own actions as an agent within structured environments such as repositories. Approaches that augment training or inference with execution-level signals or test-time verification can greatly reduce errors in such settings (Copet et al., 2025; Armengol-Estapé et al., 2025; Sharma, 2024). However, these methods rely on rich symbolic structure and closed-form semantics, and do not readily generalize to domains where such scaffolding is unavailable.

## 2.5. Vision-Language Models: From Text to Multimodal

The definition of hallucination in vision-language models (VLMs) evolved in parallel to the more text-based notions of hallucination, but also converges on similar core ideas; VLM definitions have also moved from surface-level inconsistencies towards a more world-model centered definition.

In early work on VLMs, hallucination taxonomies focused on static text-image inconsistencies: classifying surface-level errors such as object or attribute mismatches (Li et al., 2023b). This simple classification did not fully capture multimodal hallucinations, and was superseded by frameworks like HallusionBench (Guan et al., 2024), which reframed the hallucination as a **knowledge conflict** between the model's *parametric priors* (language memory) and its *contextual understanding* (visual input). This yields two primary failure modes: *language hallucination*, where strong priors override visual evidence, and *visual illusion*, where the perception module fails complex interpretation. This parallels RAG settings: prioritizing the retrieved context and parametric knowledge as sources of truth. Multimodal grounding approaches such as Feng et al. (2022) show that supplementing parametric language knowledge with visual context can reduce commonsense violations in text generation, highlighting how hallucinations can arise from unresolved conflicts between internal priors and external world evidence.

This shift deepened further with the introduction of multi-sensory and dynamic tasks. Benchmarks such as Savvy-Bench (Chen et al., 2025) reveal failures that transcend static modality-dependent inconsistencies, and highlight that models' inability to synthesize and disambiguate signals across modalities may lie in an inability to maintain a model of coherent dynamic scenarios over time. This notion of hallucination parallels newer notions in agentic text-based settings: we are concerned that a model's generated content is not merely unfaithful with respect to some input source, but rather with respect to a plausible world which we would like the model to internalize.

## 3. A General Definition of Hallucination

While different settings such as neural machine translation, summarization, multimodal generation, and agentic settings have stressed different aspects of what constitutes hallucination (Venkit et al., 2024), the common factor is a mismatch between the model's output and what we view as true. However, depending on what we take as the ground truth, different things may be viewed as "true". In summarization, truth is usually defined by the source document, regardless of facts about the outside world, while in open-domain QA, we are usually interested in real-world facts regardless of whether or not we can find some supporting document for what we generate. In VLMs, truth may be defined by visual evidence, while in agentic settings, truth may be defined by the observable state of the agent's environment. Despite these differences, the underlying structure is consistent: we want a model's definitions to always be in alignment with some underlying truth which can be derived from different sources.

In order to formalize this notion, we introduce the notion of a *reference world model* (Definition 1), which is a formal representation that captures what is objectively true in a given context. Typically, a "world model" in the literature refers to an agent's internal learned representation of its environment, which can be incomplete or incorrect. We define the reference world model to be the gold standard for what we want the model's internal world model to be.

**Definition 1** (Reference world model). *A reference world model is a tuple $W = (\mathcal{S}, \mathcal{H}, \mathcal{R})$, where $\mathcal{S}$ is a set of possible world states, $\mathcal{H}$ is a set of possible interaction histories (e.g., instructions, dialogue, logs), and $\mathcal{R}$ is a set of rules constraining which $(s, h) \in \mathcal{S} \times \mathcal{H}$ are admissible.*[1]

*For a given input $x$ and world model $W$, we assume:*

- *a view function $V$ that selects the portion of the world that is relevant for $x$:*

$$V(W, x) \subseteq \mathcal{S} \times \mathcal{H},$$

- *a conflict resolution policy $P$ that specifies how to reconcile multiple sources of information within $V(W, x)$ (e.g., "KB overrides in-context text"), and*
- *a truth function*

$$T_{W,P}(x, c) \in \{\text{true}, \text{false}, \text{unknown}\}$$

---

[1] Note that $\mathcal{R}$ is a bit of syntactic sugar here; it can also be folded into the definition such that we already work with only admissible states for each task.

*that assigns a truth status to any atomic claim $c$ given the world model $W$, input $x$, and policy $P$.*

**Definition 2** (Hallucination as inaccurate world modeling). *Let $x$ be an input and let a language model produce an output $y$ in response to $x$. Let $C(y)$ denote the set of atomic claims expressed in $y$ (e.g., factual assertions about entities, events, states of the world) that are observable to the user.*

*Given a reference world model $W$, conflict policy $P$, and truth function $T_{W,P}$ as above, we say that $y$ hallucinates with respect to $(W, P)$ if and only if $\exists c \in C(y)$ such that $T_{W,P}(x, c) =$ false. Intuitively, hallucination occurs when the world implicitly described by the model's output $y$ disagrees with the reference world model $W$ on at least one observable claim.*

**What counts as a reference world model?** We use "world model" in a minimal, functional sense. Concretely, $\mathcal{W}$ may define valid states $\mathcal{S}$, transition or consistency rules $\mathcal{R}$, and the information needed by the truth function $T_{\mathcal{W},P}$ to evaluate whether a claim is true, false, or unknown under a conflict policy $P$. Importantly, $\mathcal{W}$ is not tied to a particular representation or architecture: it may be instantiated as a simulator, database, state machine, formal grammar, knowledge base, source document, or executable program. This distinguishes $\mathcal{W}$ from a vague notion of "knowledge": it is the specified structure against which claims are evaluated.

**Why not just compare to ground truth?** A fixed ground-truth label is a special case of our framework, where the reference world $W$ is static and the truth function reduces to checking membership in an answer set. However, many hallucination settings require more structure. In agentic or partially observed environments, truth may depend on state, history, and the model's available view. For example, whether the claim "the agent can see object $X$" is true may depend on prior actions, such as whether the agent acquired a lamp or opened a door. A static ground-truth answer cannot capture this conditional dependence.

**The role of claim extraction.** Our definition assumes that a claim $y$ can be mapped to a set of evaluable claims $C(y)$, but how this is done is a design decision to be made by those creating evaluations. This challenge is not specific to our framework: any hallucination evaluation over free-form outputs must decide which claims are being judged.

Another consideration is that claim extraction should preserve epistemic status. For example, "$X$ is true" and "$X$ is likely true" should not be collapsed into the same claim. A statement such as "$X$ is likely true" would be treated as a higher-order claim about the probability of $X$ in our framework, but this example would not fall under the scope of our framework unless the model gives a falsifiable statement, such as "$X$ is 70% likely to be true". Similarly, if

a claim cannot be evaluated, it should be marked as unknown or ambiguous rather than automatically treated as hallucination. Thus, the role of $C(y)$ is not to solve all ambiguity in natural-language evaluation, but to make the unit of evaluation explicit and inspectable.

### 3.1. Examples of Hallucinations Based on Our Definition

To illustrate how our framework unifies different notions of hallucination, we present four representative cases from different domains (two are shown here, and the remaining two can be found in Appendix B). In each example, we explicitly specify the reference world model $\mathcal{W}$, the view function $\mathcal{V}$, and a conflict policy $\mathcal{P}$ to show where a hallucination manifests via the truth function $T_{W,P}$.

#### 3.1.1. EXAMPLE 1: RETRIEVAL-AUGMENTED GENERATION (RAG)

**Task:** Answer a question using retrieved documents.

**Input ($x$):** "Freedonia lies on the east coast, above New York. What is the capital of Freedonia?"

**Retrieved context:** Document $D_1$ (Wikipedia, 2020): "Freedonia is a fictional country appearing in the 1933 Marx Brothers film *Duck Soup*. As a fictional entity, it has no real capital city."

**Model output ($y$):** "The capital of Freedonia is Freedstadt, located on the eastern coast. Freedonia has a population of two million people."

**Reference world model $\mathcal{W}$:**

- $S$: Entire retrieval corpus + general world knowledge.
- $H$: The context: question and retrieved documents.
- $R$: Retrieved documents are factually correct; Freedonia is fictional with no capital.

**View function $V(\mathcal{W}, x)$:** The retrieved document $D_1$ is visible; remaining knowledge is implicit.

**Conflict policy $\mathcal{P}$:** Retrieved documents override parametric knowledge and other contextual information.

**Truth function $T_{\mathcal{W},\mathcal{P}}$:**

- Claim $c_1$: "The capital of Freedonia is Freedstadt" → **false**.
- Claim $c_2$: "Freedstadt is located on the eastern coast" → **false**.
- Claim $c_3$: "Freedonia has a population of two million people" → **unknown**. (Freedonia is fictional, but its population in the fictional world is unknown)

**Conclusion:** The output contradicts the retrieved evidence and wrongfully attributes truth to the input context when the conflict policy is defined to prioritize the evidence. Hence,

the model hallucinates. Although the third claim about Freedonia is irrelevant, its truth status is unknown unless a document can be retrieved validating its population size.

### 3.1.2. EXAMPLE 2: AGENTIC WEB NAVIGATION AKA BROWSER USE

**Task:** An agent is instructed: "Find and click the 'Submit' button on the payment form." **Environment state (current DOM):**

```
<form id="payment-form">
  <input type="text" name="card-number" />
  <input type="text" name="cvv" />
  <button id="confirm-btn">Confirm
      Payment</button>
  <button id="cancel-btn">Cancel</button>
</form>
```

**Model output ($y$):** Action: `click(button#submit-btn)`. Reasoning: "I will now click the Submit button to complete the payment."

**Reference world model $\mathcal{W}$:**

- $S$: DOM structure of the current webpage.
- $H$: Task instructions + interaction history.
- $R$: The set of interactive elements is exactly that specified by the DOM.

**View function $V(\mathcal{W}, x)$:** The agent receives a textual/structured DOM.

**Conflict policy $\mathcal{P}$:** The DOM is ground truth; instructions cannot override reality.

**Truth function $T_{\mathcal{W},\mathcal{P}}$:**

- Claim $c_1$: "A button with id 'submit-btn' exists" $\rightarrow$ **false**.
- Claim $c_2$: "There's a Submit button available" $\rightarrow$ **false**.

**Conclusion:** The agent hallucinated an element that does not exist.

**Insight:** In agentic settings we distinguish the errors below:

- **Hallucination:** Incorrect belief about environment state.
- **Planning:** Correct beliefs, incorrect action.
- **Instruction-following:** Correct beliefs, goal ignored.

### 3.1.3. TAKEAWAYS FROM EXAMPLES

Across all settings, hallucination is detected when a claim $c$ in the output satisfies $T_{\mathcal{W},\mathcal{P}}(x, c) = $ false. The diversity of hallucination is entirely self-contained by what constitutes the reference world, visible inputs, and conflict policy.

## 4. Utility of the Definition

A reader may ponder: why introduce another definition on top of the pre-existing ones—are we not introducing more baggage on top of an already overloaded term? We argue that the problem is not that we lack a good definition of hallucination, but that we have too many that capture different facets. While there exist surveys on hallucination such as Ji et al. (2023); Huang et al. (2025), to the best of our knowledge, there is limited work that attempts to *effectively* unify hallucination under a single definition. Fang et al. (2024) attempt this through a mechanism-oriented perspective, which is valuable for diagnosing why hallucinations occur and where mitigations should intervene. Our goal is complementary: to formalize what counts as hallucination in the first place so that benchmarks and claims are comparable across tasks and observability regimes.

**Making assumptions explicit.** We do not claim to be the first to observe that hallucinations reflect a mismatch with reality. What we do claim is that most current definitions *implicitly* assume some source of truth without spelling out what that source is, how it is accessed, and how conflicts are resolved. Our framework forces these assumptions out into the open. In our view, any hallucination definition must implicitly specify: (i) a reference world model $W$ that encodes what counts as true or false for the task; (ii) a view function $V$ that determines which parts of $W$ the model is supposed to have access to for a given input $x$; and (iii) a conflict policy and truth function $T_{W,P}$ that turn potentially inconsistent evidence into categorical labels (*true*, *false*, *unknown*) for atomic claims. Much of the confusion in the literature stems from leaving $W$, $V$, and $P/T$ underspecified. Our definition of hallucination as *inaccurate world modeling that is visible by the user* is, in essence, a statement that one should not claim hallucination without first specifying what $W$ and the accompanying two elements are.

**Guiding benchmark design.** A second motivation is practical: the world-model view gives a clean design space for hallucination benchmarks. Under our framework, any benchmark must explicitly answer:

- What is the reference world model $W$? E.g., a source document, a collection of documents, a knowledge base, an environment state and its history, or some combination, etc.
- What is the view $V(W, x)$ made available to the model? E.g., the full document, a retrieved subset, a snapshot of a web page, partial observations in an environment.
- How are conflicts resolved, and what is the truth function $T_{W,P}$? E.g., "document overrides KB," "DOM snapshot is ground truth for the visible page," or "unknown facts must be treated as such."

| Example | $\mathcal{W}$ (Reference World) | $V$ (Visible Inputs) | $\mathcal{P}$ (Conflict Policy) | Hallucination Type |
|---------|------------------------------|---------------------|--------------------------------|-------------------|
| Summarization | Source document | Full document | Document is truth | Intrinsic contradiction |
| Open QA | Real-world facts | Parametric memory only | Real world is truth | Incorrect factual claim |
| RAG | Entire retrieval corpus + world knowledge | Retrieved docs (+ memory) | Docs override memory | Retrieved context contradiction |
| Agentic | Environment DOM | DOM observation | Environment is truth | Observation hallucination |

*Table 1.* Comparison of hallucination types under our unified framework. Each type arises from different choices of the reference world model $\mathcal{W}$, view function $V$, and conflict policy $\mathcal{P}$.

Once $W$, $V$, and $T_{W,P}$ are explicit, hallucination evaluation becomes straightforward: an output is hallucinated if and only if it implies at least one observable claim $c$ with $T_{W,P}(x, c) = \text{false}$. This perspective brings together settings that are currently treated disparately—e.g., summarization, open-domain QA, RAG, and agentic decision-making—all under the aegis of a common evaluation template. It also motivates our proposed benchmark direction. By working with synthetic but fully specified worlds, we can construct tasks where $W$ is programmatically known, $V$ is precisely controlled, and $T_{W,P}$ can be computed exactly. This enables large-scale, language-level hallucination benchmarks in which labels are defined by construction, rather than via another LLM or human annotator.

**Clarifying what is, and is not, hallucination.** A third motivation is conceptual. In current usage, "hallucination" often collapses several distinct failure modes:

- *World-modeling errors*: the model's internal picture of the world is simply wrong.
- *Planning errors*: the model has a basically correct picture of the world but chooses a poor plan or action.
- *Incentive or reward errors*: the model may know it is uncertain but has been trained or prompted to produce confident-sounding answers regardless.

Our definition intentionally targets only the first category. We say that an output hallucinates when the implied world in the model's text or actions contradicts $W$; we do not call every wrong answer a hallucination. Errors can arise from label noise, distribution shift, adversarial perturbations, or misaligned incentives. Hallucination, in our sense, is the subset of errors that correspond to incorrect beliefs about the reference world. One way to phrase this distinction is: *error is about outputs; hallucination is about the implied world those outputs assume*. An agent that chooses a suboptimal but faithful plan is making a control or planning mistake, not hallucinating. An agent that claims to have clicked a button that does not exist, or to be on a page that the DOM shows is different, is hallucinating: its world model is wrong. Our error taxonomy closely aligns with Yann LeCun's informal 5-fold error taxonomy from mid-2025 (LeCun, 2025).

**Unifying, not renaming.** Finally, there is a social reason to be explicit. The term "hallucination" is now entrenched; attempting to ban or entirely replace it (Millidge, 2023) is unrealistic. Our goal is not to proclaim one true definition and discard existing practice, but to provide a unifying lens on what researchers are already doing. From this perspective, early faithfulness-w.r.t.-source definitions, modern factuality benchmarks, fine-grained span-level taxonomies, and agentic hallucination benchmarks can all be seen as instantiations of the same general template.

It is also possible to view existing hallucination benchmarks through our lens. Doing so reveals that benchmarks differ not only in task format or difficulty, but in how explicitly they specify the components needed to adjudicate hallucination. FavaBench (Mishra et al., 2024) is relatively well grounded because retrieval logs make the view function $V$ observable; HALoGEN (Ravichander et al., 2025) is partially grounded, with structured tasks exposing $V$ while other domains require a separately specified reference world $W$; HaluEval (Li et al., 2023a) requires more explicit claim extraction $C(y)$, and some cases remain ambiguous due to weak grounding in both $V$ and $W$, especially in multi-hop settings. This suggests that benchmark disagreements may arise not only from differences in model behavior, but from differences in how much of the evaluation world is made explicit. Our framework therefore acts as a diagnostic checklist for benchmark design: specify $W$, expose or define $V$, state $P$, and make $C(y)$ inspectable.

This diagnostic role extends naturally to mitigation work. Methods that change the information available to the model (better retrieval, richer environment state representations, multimodal grounding) are interventions on $V$; methods that change how truth is judged or expressed (external verifiers, abstention training, calibration) are interventions on $T_{W,P}$ or on incentives; architectural or training changes that improve the model's internal approximation to $W$ are world-modeling improvements. Making these targets explicit helps clarify when two methods are addressing the same problem.

In summary, our contribution is not the observation that hallucinations are "wrong" with respect to reality, but the explicit introduction of a reference world model formal-

ism and the argument that hallucination should be reserved for *inaccurate world modeling* with respect to that formalism. We believe this provides both conceptual clarity and practical guidance for the design of future hallucination benchmarks and mitigations.

# 5. Enabling the Creation of Larger-Scale Benchmarks

One practical use of viewing hallucination as world-modeling failure is that we can take advantage of the many existing synthetic or real environments which have an explicitly defined world, such as game environments or bAbI style worlds (Kuratov et al., 2024; Nematzadeh et al., 2018). As we can specify all the components of our hallucination definition in these cases, we can generate a very large number of scenarios from these environments with varying complexities in order to investigate under what circumstances models tend to hallucinate. Whenever we can specify $W_{\text{ref}}$ and a truth function $T_{W,P}(x, c)$ that evaluates atomic claims $c$ in context $x$, we can generate a large number of instances without additional human or model annotation required.

## 5.1. Case Study: Chess As a Hallucination Benchmark

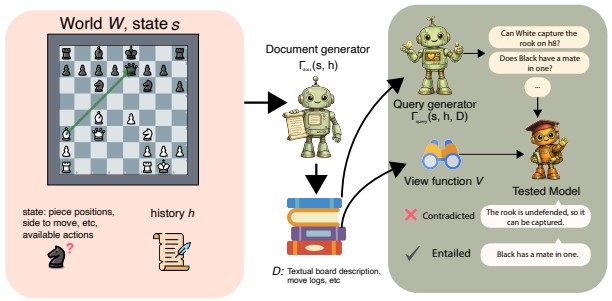

*Figure 2.* Overview of the chess scenario environment

To give a concrete example, we illustrate the benchmark creation process with the game of chess. Here, the world $W$ consists of the state space and rules describing chess, and a state $s$ is a specific board position while $h$ describes the moves that led up to that position. Figure 2 illustrates the chess environment backend as well as the construction of specific instances given a specific $(s, h)$ pair.

Given $(s, h)$, a document generator $\Gamma_{\text{docs}}$ produces textual artifacts $D = \Gamma_{\text{docs}}(s, h)$ that serve as possible context for the model. For chess, these could include a textual description of the current board (i.e. which pieces are on which squares), or a move log in a standard notation such as PGN. These documents can be directly generated from the structured state, or potentially rewritten by an LM. Next, given this, a query generator samples one or more queries $x \sim \Gamma_{\text{query}}(s, h, D)$ that probe understanding of the current

position in different ways. For instance, we could generate multiple-choice questions such as *"Does Black have a mate in one?"* or more open-ended prompts such as *"What is the best move for White in this position and why?"*. At evaluation time, a view function $V(W, x)$ determines what information about $(s, h)$ is visible to the model. In chess, this might include: (i) the current board, (ii) the board and full move history, or (iii) textual commentary on the position or similar games. By varying $V$, we can control observability and retrieval quality while keeping the underlying world $(s, h)$ fixed. For free-form generation, our claim-extraction component decomposes $y$ into a set of atomic claims $C(y)$.

Our truth function $T_{W,P}(x, c)$ then evaluates each claim $c \in C(y)$ against the reference world, using the rules in $W$ and the specific state-history pair $(s, h)$ to determine whether $c$ is entailed, contradicted, or in an unknown truth state. For chess, this would involve checking claims against the board state and legal moves. For instance, the claim *"White can capture black's queen in one move"* would be a contradiction in the state shown in Figure 3, where there is a pawn in the way. Thus, capturing the queen in one move is not possible without violating the rules of the game.

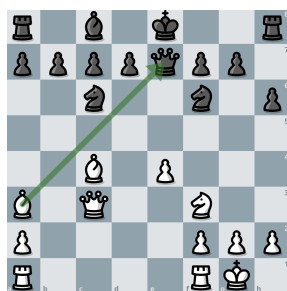 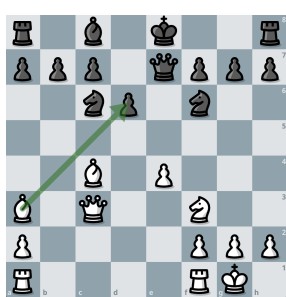

*Figure 3.* Misalignment—model's prediction (L) vs. reality (R)

Even in this simple setting, we have fine-grained control over many factors. These include the complexity of the environment (e.g., a simple endgame, where only a few pieces are on the board as opposed to a complex middlegame, with many pieces on the board and possible next moves), the observability of the environment to the model (e.g., the different documents available to the model in context), as well as the query type and answer format. Additionally, because hallucination labels are obtained automatically, we can search for settings where models tend to hallucinate more in order to increase the difficulty of the benchmark.

That said, this chess example we give is just one instantiation of our general framework. Even if LLMs eventually stopped hallucinating for chess, we can also apply our environment construction recipe to other games, such as NetHacker (Küttler et al., 2020; Piterbarg et al., 2023) or Crafter in the BAL-ROG collection (Paglieri et al., 2024). Through constructing synthetic environments or using existing challenging games, it may be possible to probe a model's propensity to halluci-

nate in almost any setting. HALLUWORLD (Liu et al., 2026) is one concrete realization of this idea spanning gridworlds, chess, and terminal tasks.

## 5.2. Beyond World-Modeling Benchmarks

It is important to note that this use of environments is conceptually distinct from existing work that probes *latent* world models or evaluates game-playing strength. Chess-based language model studies typically ask whether a model has learned an accurate internal representation of the board (Toshniwal et al., 2022; Karvonen, 2024) and can produce strong moves; metrics are Elo, legal-move accuracy, or the linear decodability of hidden states. In our framework, by contrast, the environment is used as an explicit reference world against which we evaluate *textual behavior*. The question is not whether the model's internal world model is optimal, but whether, given a specified view $V(W, x)$, its stated claims conflict with $(s, h)$. A model could have an excellent latent world model and still hallucinate by making overconfident statements under partial information, or it could have an imperfect latent model yet avoid hallucinations by abstaining when uncertain.

## 6. Alternative Views

**Hallucination is primarily about confidence calibration.** Many researchers view hallucination as primarily a problem of confidence calibration, i.e. models generating an answer even when uncertain rather than abstaining. Kalai et al. (2025) takes a computational learning theory perspective in showing that some errors during pretraining are inevitable due to available data not perfectly matching facts in the world, model capacity, and computational intractability. In their view, these errors become hallucinations because post-training on specific datasets uses binary grading of correct and incorrect, rather than incentivizing models to report *"I don't know"* when not confident. There exist works further incorporating this view into post-training objectives (Wu et al., 2025), to guide mechanistic studies (Bhatia et al., 2026), and to steer LLM inference (Yadkori et al., 2024).

**Our response:** We agree that allowing models to abstain when uncertain and improving calibration can aid hallucination mitigation, as noted in §5.2. However, calibration alone does not address errors arising from an incorrect internal world model. Consider an agent interacting with a webpage whose internal model incorrectly omits the existence of a "Submit" button. A well-calibrated agent may recognize its uncertainty and respond that it is unsure how to complete the task, thereby avoiding a hallucinated assertion. Nevertheless, the task still fails because the underlying world model is incomplete. In this sense, improved calibration can suppress hallucinated outputs without improving task performance when failures stem from missing or incorrect

representations. Our framework explicitly distinguishes confidence calibration from world modeling accuracy, highlighting that abstention alone cannot resolve failures caused by incorrect world models.

**Hallucination can be solved through retrieval or better memory access.** Another response is that hallucination is a failure on parametric memory or context. In this view, the correct knowledge exists in external sources or in the model's parameters, and the problem is through incorrect parametric knowledge, or not having access to the right external sources. This motivates several successful solutions, including retrieval-augmented generation (Lewis et al., 2020; Sun et al.; Zhang et al., 2024c; Barati et al., 2025), tools such as web search (Nakano et al., 2021), and improved attention mechanisms (Liu et al., 2024).

**Our response:** Retrieval-oriented approaches do substantially reduce hallucinations, and our framework explains this: they work by changing $V$ (the view function), providing the model with access to much more world information at inference time. This is particularly useful when $W$ consists of facts that can be found in external documents or knowledge bases. However, even with perfect retrieval, models may still misinterpret the retrieved facts or fail to resolve conflicts between facts or between facts and parametric knowledge (Xie et al., 2023; Sun et al.; Gao et al., 2025). Furthermore, retrieval does not help when information must be inferred, as in examples requiring compositional reasoning or counterfactuals (e.g. *"if gravity doubled, would birds still be able to fly?"*). Our framework clarifies when retrieval helps ($W$ is external and mostly consists of individual facts), explaining why RAG dramatically improves hallucination rates on factual QA but may not on other types of tasks.

**Hallucination in different tasks requires different engineering solutions, so a unified framework is not needed.** Lastly, some may argue that unifying summarization failures, factual QA errors, and agentic hallucinations within the same framework may not be practically helpful, since each domain has specialized ways to measure and mitigate hallucination. For instance, summarization may use faithfulness metrics and source-grounding techniques (Krishna et al., 2023; Roit et al., 2023), factual QA uses knowledge base verification and retrieval augmentation (Lin et al., 2025), while agentic systems use action validators to construct grounded reward signals (Chen et al., 2026; Gehring et al., 2024; Zhou et al., 2024; Li et al., 2025b; Zhang et al., 2024a). Since engineering solutions have progressed in each domain without a consensus on definitions, why is a unified definition necessary?

**Our response:** Domain-specific solutions can indeed be effective. However, articulating the common structure behind hallucinations allows us to develop a more scientific

view of why and where hallucinations occur. Building a common framework allows us to predict where failures will occur and why certain solutions may work in some domains but not others. For example, it explains why RAG dramatically reduces hallucinations in factual QA (where $W$ is composed of external documents) but barely helps in code execution tasks (where $W$ is program state) or agentic navigation (where $W$ is environment dynamics). It clarifies why a model might improve on summarization faithfulness benchmarks yet worsen on factual accuracy, because these measure different things (different choices of $W$).

## 7. Limitations

Our framework does not eliminate all choices in hallucination evaluation. As discussed above, claim extraction $C(y)$ remains a benchmark-design choice, and hedged, probabilistic, or unknown claims may require separate rules for epistemic status, calibration, or ambiguity. More broadly, open-domain settings such as open-domain QA do not come with a single complete reference world model. It is an engineering challenge to construct $W$, which should usually be understood as a bounded, task-specific, and versioned reference source, such as a Wikipedia snapshot, curated knowledge base, source-document collection, or clinical guidelines. This shifts the goal from judging absolute truth to judging referential consistency with respect to an explicitly specified $W$.

Conflict policies $P$ may also be subjective when sources disagree. We do not claim that every task admits a unique source-precedence rule; rather, hallucination evaluations already make such choices implicitly, and our framework requires them to be stated explicitly so that benchmark labels are interpretable and comparable.

Finally, our framework should not be read as claiming that world models are a panacea for hallucination mitigation. Retrieval, tool use, verification, calibration, and abstention may all reduce hallucinations by changing the model's accessible information, truth-checking procedure, or output policy. Our claim is instead diagnostic: hallucination evaluations and mitigations depend on assumptions about $C(y)$, $W$, $V$, and $P$, and making these assumptions explicit clarifies what failure mode a benchmark or method addresses.

## 8. Call to Action

A unified definition of hallucination makes previously intractable and disconnected problems tractable. By taking this seriously and stress-testing each component of a system's abilities with respect to hallucinations, we can compare benchmarks across tasks, understand why mitigation strategies succeed or fail, and build much larger-scale evaluations with truth determined by construction rather than

annotation. This section outlines concrete steps to realize these possibilities.

**For benchmark designers:** Specify $(W, V, P)$ when defining your evaluation. State what your reference world is, what the model observes, and how conflicts get resolved. This makes your truth function reproducible and enables meaningful comparison across benchmarks.

**For LLM application developers:** Clarify what counts as your reference world in your system. Your internal database? Retrieved documents? Live environment state? This choice, together with conflict policy, determines what counts as hallucination rather than another kind of error.

**For researchers working with new environments:** Settings where $W$, $V$, and $P$ can be precisely specified and systematically varied—such as games, simulators, formal systems, and structured databases—offer useful testbeds. Generate evaluation instances at scale and study how model behavior changes as you vary $V$ (what the model observes from $W$) or $P$ (how conflicts resolve). Particularly underexplored are scenarios where $W$ changes over time or where complex conflict policies must be inferred rather than stated.

**For mitigation work:** Different interventions primarily target different components. Retrieval and grounding methods change what the model observes ($V$). Calibration and abstention training address confidence rather than the internal world model itself. Architectural improvements aim at the internal approximation to $W$. Understanding which component an intervention targets explains why some techniques generalize across domains while others remain task-specific.

**Open research directions:** Several questions naturally follow from this framework (more discussion about future work is in Appendix A):

*Systematic study of conflict policies.* How should models behave when sources disagree? Can we train models to follow specified policies reliably?

*Benchmarks with evolving reference worlds.* Real applications require adapting to changing knowledge, e.g., databases get updated, websites change, and environments shift. How do we design models to cope with this?

*Richer environment suites.* Chess demonstrates the approach, but we need environments spanning different failure modes: web navigation (observation hallucinations), code repositories (API hallucinations), multimodal simulators (cross-modal integration), structured databases (temporal staleness), and so forth.

*Separating belief errors from control errors.* In agentic settings, wrong outcomes can stem from incorrect beliefs (hallucination), correct beliefs with poor planning, or misaligned incentives. Evaluation should distinguish these.

## Acknowledgments

We gratefully acknowledge Modal, the National Science Foundation ACCESS Program, Lambda Labs' Research Grant Program, and NVIDIA's Academic Grant Program as well as Rebecca Qian, Anand Kannappan, Bartosz Mielczarek from Patronus AI for providing compute resources that enabled this work. EL was supported by the National Sciences and Engineering Research Council of Canada (NSERC), [funding reference number 578085], as well as the SoftBank-ARM Fellowship.

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

## A. Future Work

Our framework opens up a large design space for new hallucination benchmark design and we highlight several directions that we believe are especially important.

**(1) Testing conflict policies $P$.** Conflict resolution is often the hidden source of disagreement across hallucination papers, e.g., *Does the source document override parametric memory? Does a retrieved snippet override the prompt? Does an environment observation override instructions?*, and so forth. Our framework makes $P$ explicit. The next step is to 1) systematically vary $P$ in benchmark suites, and 2) study whether models can be trained to follow a given policy reliably. This includes adversarial tests where sources conflict or are partially corrupted, and the correct behavior will need to depend on the stated policy. Further, we could assess scenarios where the model may need to correctly infer or discover the conflict policy itself, either by interacting with the user or through logical assumptions.

Note that there already exists a significant body of work studying knowledge conflicts or related contradictions between and across model parameters as well as contextual elements, e.g., retrieval augmented sources, system prompt, etc. (Wang et al., 2023; Cheng et al., 2024; Cattan et al., 2025; Liu et al., 2023). A subthread of work also studies settings with language subsets that are inclined to surface such conflicts, such as hypothetical statements (Basmov et al., 2024) and counterfactuals (Yamin et al., 2026). Naturally, a consequent subthread of work studies ways of mitigating downsides of such knowledge conflicts (Bi et al., 2025; Huang et al., 2024; Zhang et al., 2025c; Zhou et al., 2025; Zhang et al., 2025b; Pham et al., 2024).

To the best of our knowledge, we are the first to explicitly recommend stating conflict policy directly available to the LLM as a pre-condition for soundly studying hallucination detection and its mitigation in a fair, well-defined fashion.

**(2) Handling a changing (reference) world $W$.** In real world settings, there arise many situations where a language model needs to significantly adapt its world knowledge, and consequently its own internal world model, to align with an ever-changing reference world model $W$. More complicated scenarios could introduce new information or alter existing information in $W$, potentially even non-stationarily during a long-horizon interaction as the history (context) $h$ gets increasingly long. Designing hallucination benchmarks that can appropriately assess models in such scenarios with a significantly changing $W$ represents an important next step.

We acknowledge the existing collection of related work referred to by term variants such as "knowledge editing" (Zhang et al., 2024b; 2025a; Li et al., 2025a), "knowledge updating" (Ni et al., 2024; He et al., 2025; Yu and Ji,

2024; Zhang et al., 2025e), or more broadly, "temporal evolution/generalization" (Zhu et al., 2024; Tang et al., 2025). These threads of work introduce evaluation benchmarks/settings and mitigation methods for how LLMs can adapt to changed world knowledge beyond what is already baked into its parameters due to training on data until a cutoff date.

Nonetheless, our work is the first to propose unifying these threads under the wider umbrella of hallucination detection work by making the reference world $W$ a first-class citizen element of our definition, and hence of resultant benchmarks and settings. This will also enable cross-utilization of mitigation methods in newer hallucination scenarios with a changing $W$ such as non-stationary environments (Mao and Zhang, 2025). Further, one could design challenging settings with co-variation of both $W$ and $P$; for example, with a changing $W$ and a non-trivial conflict policy $P$ that admits changes in some aspects but blocks them on others. Examples include temporally complex notions such as *status quo ante* (Barale et al., 2025) and rescission (Brooks and Stremitzer, 2011) in legal settings.

**(3) Benchmark families beyond chess: richer worlds, longer histories, partial observability.** Chess is a clean base case, but future work should broaden the environment suite to cover failure modes closer to deployment:

- **Web/DOM worlds:** dynamic pages, A/B variants, and tool feedback loops (agentic observation hallucinations).
- **Codebases/Repos:** compilation/test outcomes and versioned file states as $W$ (hallucinated APIs (Spracklen et al., 2025), non-existent files, incorrect build claims).
- **Databases/logs:** structured enterprise records with controlled access patterns (RAG-style conflicts and temporal staleness).
- **Multimodal simulators:** audio-visual or embodied settings where $W$ is multi-sensory and the main challenge is cross-modal state integration.

**(4) Agentic settings: separating belief errors from control errors in interactive loops.** In multi-turn environments, wrong outcomes can arise from 1) incorrect beliefs (hallucination under our definition), 2) correct beliefs but poor planning, or 3) misaligned incentives to appear confident. A next step is to build interactive benchmarks where $W$ includes both state and execution traces, and where the evaluation separately scores claim-level state consistency, action validity, and task success. This helps prevent the term "hallucination" from becoming shorthand for *any* form of agent failure.

The term hallucination is unlikely to disappear. Our aim is to make its use both precise and encompassing enough that results are comparable across tasks and settings, while still

being actionable enough to construct benchmarks which can drive real improvements. Perfect world modeling is likely not attainable for LLMs, nor should it be the goal. Even humans have incomplete and imperfect representations of reality, especially in fields they are not adept in. However, the goal is not to become an omniscient expert, but to rather recognize the boundaries of one's knowledge, abstain when not confident, take into account external information, and keep updating one's knowledge over time. This is a worthy goal for both humans and language models.

Our framework aligns with this goal by unifying hallucination definitions from different domains and decomposing hallucination into constituent components. Importantly, we also view the proposed benchmark family as a concrete path toward measuring, diagnosing, and ultimately reducing observable world-model errors in model behavior. As language models become more capable and are deployed in higher-risk settings, systematic understanding of when language models hallucinate becomes increasingly important. Our hope is by making explicit what has been implicit previously, we can accelerate progress towards further understanding why hallucinations occur.

## B. Further Examples of Hallucinations Based on Our Definition

### B.1. Example 3: Document Summarization

**Task:** Summarize a news article about a company's quarterly earnings.

**Input article ($x$):** "TechCorp reported Q3 revenue of $2.1 billion, missing analyst expectations of $2.3 billion. CEO Jane Smith attributed the shortfall to supply chain disruptions in Southeast Asia. The company's stock fell 8% in after-hours trading."

**Model output ($y$):** "TechCorp exceeded analyst expectations with Q3 revenue of $2.1 billion. CEO Jane Smith praised the company's strong performance despite supply chain challenges."

**Reference world model $\mathcal{W}$:**

- $S$ : Set of facts stated in source document.
- $H$ : The instruction "Summarize the following article."
- $R$ : The source article is a complete and authoritative description of the events.

**View function $V(\mathcal{W}, x)$:** The complete source article is visible to the model.

**Conflict policy $\mathcal{P}$:** The source document is the single source of truth, and no external knowledge should override it.

**Truth function $T_{\mathcal{W}, \mathcal{P}}$:**

- Claim $c_1$: "TechCorp reported Q3 revenue of \$2.1 billion" $\rightarrow$ **true**.
- Claim $c_2$: "TechCorp exceeded analyst expectations" $\rightarrow$ **false** (contradicts "missing analyst expectations").
- Claim $c_3$: "CEO Jane Smith praised the company's strong performance" $\rightarrow$ **false** (contradicts attribution of "shortfall").

**Conclusion:** The output $y$ hallucinates because there exists some claim $c$ such that $T_{\mathcal{W},\mathcal{P}}(x, c) =$ false. Specifically, $c_2$ and $c_3$ are intrinsic hallucinations.

### B.2. Example 4: Open-Domain Question Answering

**Task:** Answer a factual question without explicit source documents.

**Input ($x$):** "Who won the Nobel Prize in Literature in 2023?"

**Model output ($y$):** "Haruki Murakami won the Nobel Prize in Literature in 2023 for his masterful novels exploring themes of alienation and surrealism."

**Reference world model $\mathcal{W}$:**

- $S$: The relevent envelope of real-world facts from authoritative sources (Nobel announcements, news archives).
- $H$: The question.
- $R$: The official Nobel prize records uniquely determine the winner(s) for each year.

**View function $V(\mathcal{W}, x)$:** The model has no explicit access to $\mathcal{W}$ during generation so it relies on parametric knowledge.

**Conflict policy $\mathcal{P}$:** Real-world facts override the model's parametric memory.

**Truth function $T_{\mathcal{W},\mathcal{P}}$:**

- Claim $c_1$: "Haruki Murakami won the Nobel Prize in Literature in 2023" $\rightarrow$ **false**. (the actual winner for that year is Jon Fosse)
- Claim $c_2$: "Haruki Murakami writes novels exploring themes of alienation and surrealism" $\rightarrow$ **true**.

**Conclusion:** The output hallucinates because $c_1$ is false. Here $V(\mathcal{W}, x)$ is empty, but $\mathcal{W}$ still exists. The model's output still diverges from $\mathcal{W}$, leading to hallucination.

