# OpenReview forum: "Position: We Need A Unified Definition of Hallucination (It’s The World Model, Stupid!)"
_ICML.cc/2026/Position_Paper_Track — ICML 2026 Position Paper Track regular_

### Official Review · Reviewer_Duka · 2026-02-21

**Significance:** 3
**Argument Clarity:** 3
**Rating:** 4
**Confidence:** 4

**Questions:**

I would appreciate it if the authors could provide a precise definition of a "world model" as referenced in this paper. In particular, how do you distinguish it from token-based methods (e.g., web search), and do you agree with a perspective I encountered that "Research on World Models Is Not Merely Injecting World Knowledge into Specific Tasks"? I am very interested in this direction and look forward to discussing this with you.

**Alternative Views Section:**

Yes

**Compliance With Llm Reviewing Policy A Conservative:**

Affirmed.

**Discussion Potential:**

3

**Paper Summary:**

This position paper centers on the relationship between **world models** and LLM hallucinations, advocating that world models are a core solution to address hallucination issues. It challenges the ambiguity in existing hallucination definitions and argues for aligning anti-hallucination efforts with world model research. The paper discusses potential pathways such as retrieval and memory access optimization, and attempts to frame hallucination mitigation within the scope of world model advancement, aiming to establish a clearer connection between architectural design and hallucination control in LLMs.

**Position:**

Yes

**Position In Title:**

Yes

**Related Work:**

3

**Strengths And Weaknesses:**

### Strengths
1. The paper addresses a critical and timely topic, as hallucination remains a major bottleneck for LLM deployment.
2. The emphasis on clarifying definitions (especially for hallucination) is a valuable and necessary starting point for rigorous research in this area.
3. The position is well-motivated and has the potential to stimulate meaningful discussion on the theoretical underpinnings of anti-hallucination techniques.

### Weaknesses
1. While the paper critiques the ambiguity of "hallucination" definitions, I argue that existing research related to world models has yet to provide a sufficiently clear or bounded definition of a **world model** itself.
2. The paper overstates the scope of world models by implying they can encompass all types of hallucinations. World models, as a perception-core paradigm to overcome token-based limitations, should be distinguished from token-based solutions like **web search**. Many web search-related tasks only require text processing to mitigate hallucinations, and such text-centric scenarios do not rely on the perception-core characteristics of world models.
3. As web search is a primary source of hallucination mitigation for factual errors but, in my view, falls outside the world model paradigm, the core position that world models are the panacea for hallucinations is weakened due to this definitional boundary issue.

**Support:**

3

---

> ### Author Rebuttal · Authors · 2026-03-29
>
> Thanks for the supportive comments on what to discuss and improve on.
>
> **W1: (…) I argue that existing research related to world models has yet to provide a sufficiently clear or bounded definition of a world model itself.**
>
> We agree that the term “world model” is often overloaded. In our paper, we use it in a minimal, functional (yet general) sense: a system that defines (i) valid states $\mathcal{S}$, (ii) transition or consistency rules $\mathcal{R}$, and (iii) a mechanism for evaluating claims against these (i.e., the truth function $T_{\mathcal{W},P}$). This can be applied to many settings, including webpages and actions [1], Minecraft [3] or pixels, etc.
>
> Importantly, $\mathcal{W}$ is not tied to a specific representation or architecture. It can be instantiated as a simulator, database, state machine, formal grammar, or executable Python code [2] - any structure that specifies what is true in a given setting.
> This clarification distinguishes $\mathcal{W}$ from vague notions of “knowledge” and grounds it as a verifiable reference used for evaluation. We’ll make this definition more explicit.
>
> **W2: The paper overstates the scope of world models by implying they can encompass all types of hallucinations (…) such text-centric scenarios do not rely on the perception-core characteristics of world models.**
>
> We thank the reviewer for this distinction. We do not claim that world models are limited to perception-heavy or pixel-based settings, nor that they encompass all mitigation methods.
>
> In our framework, a world model $\mathcal{W}$ is defined structurally - a system specifying valid states and transition or consistency rules - rather than by the modality of its observations. This allows $\mathcal{W}$ to apply equally to continuous environments (e.g., vision-based) and discrete, text-based settings (e.g., web navigation).
>
> For instance, Web World Models [1] demonstrate that navigating the internet is governed by a discrete transition logic that is effectively a world model in token-space, and can thus be modeled within the same $(\mathcal{W}, V, P, T)$ formalism.
> We’ll clarify that our notion of world models is structure-centric rather than modality-specific, and does not exclude text-centric scenarios.
>
> **W3: As web search is a primary source of hallucination mitigation for factual errors (…) the core position that world models are the panacea for hallucinations is weakened (…)**
>
> We do not claim that world models are the mechanism by which hallucinations must be mitigated. Rather, that a construct like a reference world model $\mathcal{W}$ is necessary for *defining and evaluating* hallucination, independent of how mitigation is achieved.
>
> In particular, tools such as web search improve the model’s access to information (i.e., they modify $V$), but do not themselves define what is true. The role of $\mathcal{W}$ is to provide the underlying standard against which claims are judged, regardless of whether the model uses retrieval, tools, or parametric knowledge to produce them.
>
> Thus, web search and similar methods are complementary to our framework: they help reduce hallucinations in practice, while $\mathcal{W}$ specifies what counts as hallucination in the first place. We’ll make this distinction more explicit.
>
> **Q1: I would appreciate it if the authors could provide a precise definition of a "world model" as referenced in this paper. In particular, how do you distinguish it from token-based methods (…)**
>
> We thank the reviewer for this thoughtful question. We define a world model $\mathcal{W}$ as a functional transition system consisting of a state space $\mathcal{S}$ and transition dynamics $\mathcal{T}$ that govern how states evolve and which claims are valid within the environment [1, 2]. The action space $\mathcal{A}$ may consist of tokens, pixels, or other modalities.
>
> We distinguish this from token-based methods such as web search via a structural vs. instrumental lens: web search is an *action* $a \in \mathcal{A}$ used to retrieve info, whereas $\mathcal{W}$ defines the underlying structure (or “physics”) that determines whether a claim $c$ is consistent with the environment.
>
> We concur with the view of 'Research on World Models is not merely injecting world knowledge into specific tasks.' [4]. While knowledge injection improves coverage, world models capture the *causal and structural constraints* that govern valid states and transitions, enabling consistent reasoning across contexts [4,5].
>
> In our framework, $\mathcal{W}$ serves as the invariant evaluative backbone against which claims are judged, independent of the tools or methods used to produce them. We’ll clarify this definition further.
>
> [1] Web World Models (arxiv:2512.23676)
>
> [2] Generating Code World Models Guided by LLMs with MCTS (arxiv:2405.15383)
>
> [3] MineWorld (arxiv:2504.08388)
>
> [4] Research on World Models Is Not Merely Injecting World Knowledge into Specific Tasks (arxiv:2602.01630)
>
> [5] World of Workflows (arxiv:2601.22130)

---

> > ### Author Rebuttal · Reviewer_Duka · 2026-04-02
> >
> > Thank you for your response. There is one point I must discuss with you. First, I respect everyone’s definition of world models. However, from the perspective that the goal of world models is to enable pre-trained models (trained on internet data) to truly handle complex physical world tasks, methods such as web world models and code world models actually fall outside the scope of problems that world models need to solve. Of course, different researchers have different views on the definition of world models. As a researcher in areas related to world models, although I do not recognize web world models or code world models as valid world model research, I am not as narrow-minded as some other researchers in this field who insist on following their definitions. I find this paper interesting, and therefore I have decided to keep my score as 4.

---

### Official Review · Reviewer_hcUp · 2026-03-08

**Significance:** 2
**Argument Clarity:** 4
**Rating:** 4
**Confidence:** 3

**Questions:**

This point is more of a suggestion than a question. The one important aspect that should not be overlooked is that hallucinations typically appear to be correct or well-grounded outputs but are in fact incorrect. Without capturing this property, it becomes difficult to distinguish a generic incorrect answer from a hallucinated one. Adding an additional criterion to reflect this “plausible but false” characteristic could make the definition more complete and better aligned with the common understanding of hallucination.

**Alternative Views Section:**

Yes

**Compliance With Llm Reviewing Policy A Conservative:**

Affirmed.

**Discussion Potential:**

3

**Final Justification:**

The rebuttal has addressed my concerns. I've updated my score.

**Paper Summary:**

The main position of this paper is that hallucination should be defined in a unified way by comparing model outputs to a reference world model. If the model produces an output that is incorrect with respect to the world model, then it is hallucinating. The paper uses four example domains: summarization, open-domain question answering, retrieval-augmented generation, and agentic tasks, to demonstrate this unified definition. It also shows how the definition can be used to construct benchmarks for examining hallucinations, using chess modeling as an example. Finally, it presents three alternate views in contrast with the proposed position.

**Position:**

Yes

**Position In Title:**

Yes

**Related Work:**

3

**Strengths And Weaknesses:**

Strengths:
1. The proposed definition is intended to be both more fine-grained and more broadly applicable than previous definitions. Before evaluating whether a model is hallucinating, it essentially asks three questions: What is the reference world model for the large language model? What information does it have access to? How does the model resolve conflicts between different sources of information? This viewpoint captures finer details of hallucination.
2. The proposed definition also unifies different domains. The paper provides four clear examples showing how the definition applies across settings. Using the same general framework of world models, view functions, and truth functions, it becomes possible to discuss these different domains through a single lens.


Weaknesses:

1. The paper does not clearly explain the advantages of using a reference world model over simply comparing model outputs to a ground truth, as in prior definitions. Providing a concrete example where a straightforward ground-truth comparison fails, but the proposed definition successfully captures the error, would help clarify the benefit of this approach.

3. The proposed definition is too general. For example, in the retrieval-augmented generation example in Section 3.1.1, the model could respond with something like “A4873290dasd” to the question “What is the capital of Freedonia?” and this would still be classified as a hallucination. The paper argues that not all incorrect outputs are hallucinations (in Section 4 Clarifying what is, and is not, hallucination section). It implies that some errors, such as label noise or distribution shift, are not hallucination-related. However, under the proposed definition, these errors also stem from an incorrect world model: label noise can cause the model to treat incorrect outputs as true, and distribution shift implies that the model learned an inaccurate representation during training. Yet the paper claims these cases fall outside hallucination. This suggests that, in practice, the proposed definition ends up treating all incorrect outputs as hallucinations. If such is the case, does it make sense to use the proposed definition for describing hallucinations?

3. In the alternative views section, View 1 and View 2 do not really read as alternative views on the definition. The paper argues for a unified definition of hallucinations and explains why such a definition is necessary. However, View 1 frames hallucination as an issue of confidence calibration, and View 2 suggests that hallucination can be addressed through retrieval. Both points focus on methods for mitigating hallucinations rather than offering alternative definitions. Since the paper’s contribution is primarily definitional, the alternative views section would be stronger if it presented alternative definitions of hallucination or debated the necessity of a unified definition, rather than proposing different mitigation approaches.

**Support:**

2

---

> ### Author Rebuttal · Authors · 2026-03-29
>
> We thank you for perceptive comments. We respond to each below.
>
> **W1: The paper does not clearly explain the advantages of using a reference world model over simply comparing model outputs to a ground truth (…)**
>
> Thanks for the suggestion. The advantage of a reference world model over fixed ground-truth (GT) comparison is that it supports *stateful, conditional truth*, which cannot be captured by a static answer set.
>
> In many settings (especially agentic ones), correctness depends on interaction history. For example, in a partially observed environment, acquiring an item (e.g., a lamp) may change what the agent can see. Whether a claim like “the agent can see object X” is true depends on prior actions, which a fixed GT label cannot capture.
>
> Thus, “ground truth” becomes a function of state and history, requiring a structured representation, which we formalize as a reference world model $\mathcal{W}$.
>
> We’ll clarify this distinction and add a concrete example where static GT comparison fails but world-model-based evaluation succeeds.
>
> **W2: The proposed definition is too general (…)**
>
> Our definition does not classify all incorrect outputs as hallucinations because it applies only to *interpretable claims* $c \in C(y)$.
> First, outputs not semantically parseable into claims (e.g., “A4873290dasd”) yield no valid $c \in C(y)$, so $T_{W,P}(x,c)$ is undefined (not false); these are not hallucinations. Unknown or undefined claims reflect epistemic uncertainty or incomplete knowledge, not hallucination (see response to Reviewer imxN W2).
>
> Second, among interpretable outputs, hallucination is a *subset* of errors: it requires $T_{W,P}(x,c) = \text{false}$. Other failures (e.g., instruction-following) may produce incorrect outputs without false claims and are thus not hallucinations.
>
> Finally, we distinguish the model’s internal beliefs from the evaluator’s reference world $\mathcal{W}$. A model may internally “believe” an incorrect fact (e.g., “the sky is green”), but it is still hallucinating relative to $\mathcal{W}$, since hallucination is defined w.r.t. external truth.
>
> We’ll clarify this and add a brief taxonomy distinguishing non-claim errors, general errors, and hallucinations.
>
> **W3: In the alternative views section, View 1 and View 2 do not really read as alternative views on the definition (…)**
>
> We’ll revise our alternative views to add *alternative definitions* of hallucination. While many works implicitly adopt different notions, few provide explicit formal definitions - this motivates our work to make such assumptions explicit in the first place. We brainstorm a few alternatives below:
>
> **a) The Epistemic View: Hallucination as Miscalibration**
> $\mathcal{H}_{\text{Epi}} \iff \text{Conf}(c) \gg \text{Acc}(c \mid W)$.
>
> This counts only overconfident falsehoods as hallucinations, treating them as failures of internal calibration rather than incorrect world modeling. It conflates hallucination with miscalibration and ignores low-confidence false claims, whereas our definition captures correctness relative to $\mathcal{W}$ independent of confidence.
>
> **b) The Relational View: Hallucination as Local Inconsistency**
>
> $\mathcal{H}_{\text{Rel}} \iff y \nvdash x$.
>
> This discards the reference world and defines truth via entailment between input $x$ and output $y$, treating the model as a closed system. It cannot detect factually incorrect but contextually consistent outputs, whereas our definition evaluates correctness against an explicit external world $\mathcal{W}$.
>
> **c) The Functionalist View: Hallucination as Task-Relevant State Error**
>
> $\mathcal{H}_{\text{Func}}$ iff $(\hat{s}_t \neq s_t)$ and $(V^* (s_t) - Q^* (s_t, \pi(\hat{s}_t)) > \epsilon)$
>
> This defines hallucination as an operationally significant state mismatch, ignoring semantically false claims that do not affect reward. In contrast, our definition provides a task-agnostic notion of correctness grounded in $\mathcal{W}$.
>
> These above definitions emphasize confidence, local consistency, or utility, rather than correctness relative to $\mathcal{W}$. We’ll add these alternative definitions to our next revision.
>
> **Q1: (…) Adding an additional criterion to reflect this “plausible but false” characteristic could make the definition more complete (...)**
>
> We thank you for this suggestion. However, we argue that plausibility is inherently *observer-dependent* and should not be part of the core definition.
>
> To avoid conflating correctness with user perception, we distinguish:
> - Evaluator: our framework $(\mathcal{W}, V, P, T)$, which determines truth relative to $\mathcal{W}$
> - Observer: a user with partial knowledge $W_{\text{reader}}$
>
> Plausibility corresponds to consistency with $W_{\text{reader}}$ but not $\mathcal{W}$; a claim may appear valid to an observer yet be false under $\mathcal{W}$.
>
> This preserves an objective definition of hallucination while allowing plausibility to be analyzed separately. We’ll clarify this and include an example.

---

> > ### Author Rebuttal · Reviewer_hcUp · 2026-04-04
> >
> > Thanks for the rebuttal. I'll update my score.

---

### Official Review · Reviewer_pzs2 · 2026-03-11

**Significance:** 2
**Argument Clarity:** 3
**Rating:** 4
**Confidence:** 3

**Questions:**

1. How can such a massive reference world model be concretely constructed and quantified in open-domain QA, where knowledge boundaries are ambiguous and absolute truth is often absent?

2. In complex scenarios with contradictory multi-source information (e.g., RAG systems processing conflicting documents), defining a conflict policy to determine source precedence remains highly subjective.

3. Beyond providing theoretical guidance for constructing synthetic benchmarks, how does this unified framework directly inspire or inform novel techniques for hallucination mitigation?

**Alternative Views Section:**

Yes

**Compliance With Llm Reviewing Policy A Conservative:**

Affirmed.

**Discussion Potential:**

3

**Final Justification:**

I have read the rebuttal and appreciate the authors' efforts. My assessment remains positive and unchanged.

**Paper Summary:**

This position paper proposes unifying the definition of hallucination in large language models as user-observable, inaccurate internal world modeling. While current definitions are highly fragmented across tasks (e.g., summarization, QA, agents), the authors consolidate them into a universal framework based on three explicit prerequisites: a reference world model representing objective facts, a view function defining the information accessible to the model, and a conflict policy for resolving contradictions. Under this framework, a hallucination occurs when a model's statement evaluates as false against the specified reference world and conflict policy. This unified approach compels researchers to clarify their evaluation assumptions, successfully isolating true hallucinations from planning or reward calibration errors.

**Position:**

Yes

**Position In Title:**

Yes

**Related Work:**

3

**Strengths And Weaknesses:**

Strengths

1. This paper consolidates the fragmented definitions of "hallucination" across subfields (e.g., summarization, open-domain QA, multimodal, and agents) into a single framework.
2. This paper prevents conceptual conflation by strictly distinguishing true hallucinations from planning and control errors, instruction-following failures, or confidence calibration issues.
3. By proposing the use of strictly rule-bound synthetic environments as reference worlds, it enables the low-cost, automated generation of large-scale, annotation-free evaluation benchmarks.

Weaknesses

1. While effective in closed, rule-based environments, defining a precise reference world and truth function remains exceedingly difficult in open-domain generation where knowledge is ambiguous and boundaries are blurred.
2. In complex real-world contexts (such as conflicting information across multi-source documents), determining source precedence is highly subjective and difficult to formalize.
3. As a position paper, its core contribution is limited to redefining and standardizing evaluation paradigms.

**Support:**

3

---

> ### Author Rebuttal · Authors · 2026-03-29
>
> Thanks for your valuable comments and feedback!
>
> **W1: (…) defining a precise reference world and truth function remains exceedingly difficult in open-domain generation (…)**
>
> We do not assume a single world model for all human knowledge. Our framework operates with a *local reference world* $\mathcal{W}$ defined relative to the task.
>
> In open-domain settings, $\mathcal{W}$ can be operationalized as a specific, versioned source (e.g., a snapshot of Wikipedia or a curated KB), with rules for extracting and evaluating facts. This makes evaluation defined and reproducible, even when the underlying world is large or evolving.
>
> While constructing $\mathcal{W}$ in open domains is challenging, we view this as an engineering and data limitation rather than a limitation of the definition. Making $\mathcal{W}$ explicit is precisely what enables us to turn an otherwise ambiguous notion of “truth” into something inspectable and comparable.
>
> We’ll clarify this open-world setting more explicitly.
>
> **W2: In complex real-world contexts (…), determining source precedence is highly subjective and difficult to formalize.**
>
> We agree that source precedence may be subjective. We do not claim that one can always derive a unique policy $P$, but that hallucination evaluations already rely on such choices implicitly. The role of our framework is to make that choice explicit. In many tasks, $P$ is natural and task-defined (e.g., source doc overrides outside knowledge in summarization). In harder settings with genuinely conflicting sources, different policies may be reasonable; precisely for that reason, benchmarks should state which policy they adopt as $P$ affects the truth function and what counts as hallucination. We’ll clarify that $P$ should not remove all subjectivity, but make evaluative assumptions interpretable and comparable.
>
> **W3: As a position paper, its core contribution is limited to redefining and standardizing evaluation paradigms.**
>
> We agree that our primary contribution is definitional. However, we argue that a shared framework is a necessary foundation for progress.
>
> Without a unified structure such as $(\mathcal{W}, V, P, T)$, it is difficult to precisely define what is being evaluated, disentangle failure modes, or compare results across benchmarks. Our framework makes assumptions explicit, enabling reproducible and interpretable evaluation.
>
> We view standardization not as an end goal, but enabling infrastructure: it shifts hallucination research from subjective annotation toward scalable evaluation and the systematic development and comparison of mitigation methods. We’ll clarify this motivation.
>
> **Q1: How can such a massive reference world model be concretely constructed and quantified in open-domain QA (…) ?**
>
> We agree that constructing a complete reference world for open-domain QA is challenging. Our framework does not require a universal $\mathcal{W}$, but instead operates with a *bounded, task-specific reference world*.
>
> In practice, $\mathcal{W}$ can be instantiated as a versioned source (e.g., a Wikipedia snapshot, or curated KB), with rules for extracting and evaluating claims. This makes evaluation defined and reproducible, even when knowledge is ambiguous or evolving.
>
> We shift the goal from defining absolute truth to evaluating *referential consistency* w.r.t. a specified $\mathcal{W}$. While constructing such $\mathcal{W}$ is an engineering challenge, making it explicit allows hallucination to be measured in a precise and comparable way. We’ll clarify this.
>
> **Q2: In complex scenarios with contradictory multi-source information (…), defining a conflict policy to determine source precedence remains highly subjective.**
>
> Like our response to W2, source precedence may be subjective. We claim that hallucination evaluations already rely on such subjective choices implicitly, and the contribution of $P$ is to make all assumptions explicit, enabling interpretable and reproducible evaluation and mitigation efforts.
>
> **Q3: (…) how does this unified framework directly inspire or inform novel techniques for hallucination mitigation?**
>
> This highlights another key contribution of our proposed definition: it guides the investigation of more mitigation techniques. Our framework decomposes hallucination into components $(\mathcal{W}, V, P, T)$, which directly map to intervention points for mitigation.
> - **Improving $V$**: retrieval, tools, multimodal grounding
> - **Improving $T$**: calibration, abstention, verification
> - **Improving $\mathcal{W}$**: better training / architectures
> - **Improving $P$**: handling conflicting sources
>
> This explains why methods behave differently across settings (e.g., RAG helps when $\mathcal{W}$ is external, but less for incorrect internal modeling). It also suggests new directions, such as learning conflict policies $P$, dynamically adapting $V$, and separating belief (hallucination) from planning errors. We’ll make this mitigation-oriented perspective more explicit.

---

> > ### Author Rebuttal · Reviewer_pzs2 · 2026-04-03
> >
> > The authors have clarified my doubts and addressed the issues I raised. I am keeping my positive score.

---

### Official Review · Reviewer_imxN · 2026-03-12

**Significance:** 3
**Argument Clarity:** 3
**Rating:** 4
**Confidence:** 4

**Questions:**

1. When a model produces a hedged claim such as "X is probably true," how is this handled in $C(y)$? Is it treated as the same claim as "X is true"?

2. What is the authors' position on unknown claims? If a model assertively states something for which $T_{W,P}(x, c) = \text{unknown}$, does this constitute hallucination under the proposed framework?

3. Have the authors attempted a post-hoc analysis of existing benchmarks (e.g., HaluEval, FavaBench, HALoGEN) through the lens of $(W, V, P)$? If so, did this reveal any previously unnoticed differences or patterns?

**Alternative Views Section:**

Yes

**Compliance With Llm Reviewing Policy A Conservative:**

Affirmed.

**Discussion Potential:**

3

**Final Justification:**

The authors have fully addressed my concerns.Therefore, I maintain my positive score.

**Paper Summary:**

This paper argues that existing definitions of hallucination across domains (summarization, QA, RAG, multimodal, agentic) are fragmented and can be unified under a single framework. The  authors propose that all definitions implicitly assume three components a reference world model $W$, a view function $V$, and a conflict policy $P$  and that hallucination is simply inaccurate world modeling: a model hallucinates iff its output implies an observable claim $c$ where $T_{W,P}(x,c) = \text{false}$. The paper shows prior definitions map to different choices of $(W, V, P)$, proposes scalable benchmarks via synthetic environments (illustrated with chess), and introduces an error taxonomy separating hallucination from planning and incentive errors.

**Position:**

Yes

**Position In Title:**

Yes

**Related Work:**

3

**Strengths And Weaknesses:**

**Strengths**

1. Important and timely problem. The fragmentation of hallucination definitions is a real barrier to cross benchmark comparison and mitigation generalization. The paper captures this clearly.

2. Intuitive and comprehensive framework.The framework naturally classifies mitigation strategies by their target: $V$ (retrieval), $T_{W,P}$ (calibration), or $W$ (architectural improvements).

3. Useful error taxonomy. The hallucination vs. planning error vs. incentive error distinction follows naturally from the formalization and is practically valuable for agentic settings.


**Weaknesses**

1. Claim extraction $C(y)$ is underspecified. The framework assumes decomposition into atomic claims, but hedged assertions, implicatures, and the fact-opinion boundary make this non-trivial. The status of these cases within the framework is unclear.

2. Unknown claims are ambiguous. When a model confidently asserts a claim where $T_{W,P}(x,c) = \text{unknown}$, it is unclear whether this constitutes hallucination. This ambiguity could significantly affect benchmark design.

3. Open-world applicability needs more candid discussion. Examples focus on closed-world or well-defined settings. The argument that "making $W$ explicit is itself valuable" is reasonable, but a franker discussion of how feasible and useful this is when $W$ is inherently ill-defined (medical QA, temporally evolving facts, etc.) would better clarify the framework's scope.

**Support:**

3

---

> ### Author Rebuttal · Authors · 2026-03-29
>
> Thanks for thorough comments!
>
> **W1: Claim extraction C(y) is underspecified (...)**
>
> We agree that claim extraction $C(y)$ is non-trivial. However, this challenge is not unique to our framework, but is inherent to all hallucination evaluation methods, which must map free-form outputs to evaluable units (e.g., facts, spans, statements). We make this step explicit via $C(y)$, rather than leaving it implicit in annotation or evaluation pipelines.
>
> Our framework does not require a unique or fully solved decomposition, but only that outputs can be mapped to *interpretable claims* $c$ for which $T_{\mathcal{W},P}(x,c)$ can be evaluated. This can be implemented using standard techniques (e.g., atomic fact extraction or entailment-based decomposition), with different choices of $C(y)$ corresponding to explicit evaluation protocols.
>
> For hedged statements (e.g., “X is likely true”), we treat these as higher-order claims about uncertainty rather than direct assertions of $X$. We’ll clarify the role of $C(y)$ and discuss these edge cases.
>
> **W2: Unknown claims are ambiguous (…)**
>
> To clarify, under our definition, hallucination is strictly tied to false claims (i.e., $T_{w,p}(x,c) = false$). Claims with $T_{w,p}(x,c) = unknown$ are not hallucinations but instead reflect epistemic uncertainty or incomplete knowledge. This distinction is intentional: it separates world-model errors from calibration or abstention failures. We’ll clarify this and explicitly distinguish these cases and suggest evaluating them as a separate axis (e.g., over-assertion under uncertainty).
>
> **W3: Open-world applicability needs more candid discussion (…)**
>
> We agree this merits more discussion. In domains e.g., medical QA or evolving knowledge, $\mathcal{W}$ may be large, incomplete, or time-dependent.
>
> Our key claim is that making $\mathcal{W}$ explicit remains useful precisely in these settings, because it introduces *diagnostic clarity*. In practice, $\mathcal{W}$ can be operationalized as a *local reference world* (e.g., a versioned clinical guideline or a snapshot of a KB), which makes evaluation well-defined.
>
> This allows us to distinguish two failure modes that are otherwise conflated:
>
> - Temporal drift: the model is consistent with $ \mathcal{W} _{t} $ but not $ \mathcal{W} _{t'} $
> - Internal Hallucination: the model contradicts the specified $\mathcal{W}$
>
> While constructing such $\mathcal{W}$ is an engineering challenge in open domains, we view this as a limitation of current systems rather than of the definition itself. Without an explicit $\mathcal{W}$, “truth” remains implicit and results are difficult to interpret or compare.
>
> We’ll more clearly discuss these open-world considerations in our next revision.
>
> **Q1: When a model produces a hedged claim such as "X is probably true," how is this handled in C(y)? (…)**
>
> We treat hedged statements (e.g., “$X$ is probably true”) not as the claim $X$, but as a higher-order claim about uncertainty (e.g., $\text{Prob}(X)$).
>
> Under our framework, hallucination is defined only for factual claims $c$ with $T_{\mathcal{W},P}(x,c) = \text{false}$. In contrast, uncertainty statements are evaluated separately: they may reflect appropriate uncertainty (e.g., when $T_{\mathcal{W},P}(x,c) = \text{unknown}$) or miscalibration (e.g., unjustified confidence), as we have touched upon in the response to W2.
>
> This distinction allows us to separate *world-model errors* (false claims) from *calibration errors* (misstated uncertainty). We’ll clarify how $C(y)$ handles such higher-order claims and their evaluation.
>
> **Q2: What is the authors' position on unknown claims? (…)**
>
> Similar to response to W2, under our framework, claims with $T_{\mathcal{W},P}(x,c) = \text{unknown}$ are *not* considered hallucinations. Hallucination is strictly tied to false claims, i.e., $T_{\mathcal{W},P}(x,c) = \text{false}$.
>
> When a model assertively states an unknown claim, this reflects a different failure mode - namely, miscalibration or overconfidence - rather than an incorrect world model. The issue is not that the model asserts a false fact, but that it fails to appropriately express uncertainty.
>
> This distinction is intentional: it separates *world-model errors* (false claims) from *epistemic/calibration errors* (overconfident unknowns). We’lll revise the paper to make this diff explicit.
>
> **Q3: Have the authors attempted a post-hoc analysis of existing benchmarks (…)**
>
> We conducted a brief post-hoc analysis. Our framework reveals systematic differences across benchmarks. At the claim level: FavaBench (\~75%) is largely verifiable via retrieval logs (V observable); HALoGEN (\~70%) is partially verifiable (structured tasks expose V, others require domain-specific W); HaluEval (\~60%) requires explicit C(y) extraction, with ~20% remaining ambiguous due to weak grounding in both V and W (especially multi-hop). These differences reflect varying degrees of observability of V and definability of W, informing benchmark design.

---

> > ### Author Rebuttal · Reviewer_imxN · 2026-04-03
> >
> > The rebuttal addresses my concerns well. The 'unknown' vs 'false distinction' (W2) is clean, the hedged claim handling (W1) is reasonable, and the post-hoc benchmark analysis (Q3) demonstrates the framework's utility.

---

### Decision · Program_Chairs · 2026-04-30

**Decision:**

Accept (regular)

**Comment:**

This paper makes a clear and well-motivated position. The reviewers are overall leaning positive. However, I have two issues: (a) The claim extraction is hazy, and (b) world model cannot be treated as a panacea. The authors need to clearly state the limitation of their position.